# A large-scale brain network mechanism for increased seizure propensity in Alzheimer's disease

**Luke Tait**[1]*, **Marinho A. Lopes**[1], **George Stothart**[2], **John Baker**[3], **Nina Kazanina**[4], **Jiaxiang Zhang**[1], **Marc Goodfellow**[5]

**1** Cardiff University Brain Research Imaging Centre, Cardiff University, Cardiff, United Kingdom, **2** Department of Psychology, University of Bath, Bath, United Kingdom, **3** Dementia Research Centre, Queen Square Institute of Neurology, UCL, London, United Kingdom, **4** School of Psychological Science, University of Bristol, Bristol, United Kingdom, **5** Living Systems Institute, University of Exeter, Exeter, United Kingdom

* TaitL2@cardiff.ac.uk

**Data Availability Statement:** Data cannot be shared publicly because of ethical constraints. Data are available from the University of Bristol Institutional Data Access Committee (contact via

## Abstract

People with Alzheimer's disease (AD) are 6-10 times more likely to develop seizures than the healthy aging population. Leading hypotheses largely consider hyperexcitability of local cortical tissue as primarily responsible for increased seizure prevalence in AD. However, in the general population of people with epilepsy, large-scale brain network organization additionally plays a role in determining seizure likelihood and phenotype. Here, we propose that alterations to large-scale brain network organization seen in AD may contribute to increased seizure likelihood. To test this hypothesis, we combine computational modelling with electrophysiological data using an approach that has proved informative in clinical epilepsy cohorts without AD. EEG was recorded from 21 people with probable AD and 26 healthy controls. At the time of EEG acquisition, all participants were free from seizures. Whole brain functional connectivity derived from source-reconstructed EEG recordings was used to build subject-specific brain network models of seizure transitions. As cortical tissue excitability was increased in the simulations, AD simulations were more likely to transition into seizures than simulations from healthy controls, suggesting an increased group-level probability of developing seizures at a future time for AD participants. We subsequently used the model to assess seizure propensity of different regions across the cortex. We found the most important regions for seizure generation were those typically burdened by amyloid-beta at the early stages of AD, as previously reported by in-vivo and post-mortem staging of amyloid plaques. Analysis of these spatial distributions also give potential insight into mechanisms of increased susceptibility to generalized (as opposed to focal) seizures in AD vs controls. This research suggests avenues for future studies testing patients with seizures, e.g. co-morbid AD/epilepsy patients, and comparisons with PET and MRI scans to relate regional seizure propensity with AD pathologies.

data request form at http://www.bristol.ac.uk/staff/researchers/data/accessing-research-data/) for researchers who meet the criteria for access to confidential data. The computational model and underlying source codes described in this publication are available freely for academic use at https://github.com/lukewtait/AlzheimersBNI.

**Funding:** This work was supported by the European Research Council [Grant Number 716321] (LT/JZ). This work was supported by the EPSRC [Grant Numbers EP/P021417/1 and EP/N014391/1] (MG); a Wellcome Trust Institutional Strategic Support Award (https://wellcome.ac.uk/) [Grant Number WT105618MA] (MG); University Research Fellowship from the University of Bristol (NK); MAL gratefully acknowledges funding from Cardiff University's Wellcome Trust Institutional Strategic Support Fund (ISSF) [Grant Number 204824/Z/16/Z]. The funders had no role in study design, data collection and analysis, decision to publish, or preparation of the manuscript.

**Competing interests:** The authors have declared that no competing interests exist.

## Author summary

People with Alzheimer's disease (AD) are more likely to develop seizures than cognitively healthy people. In this study, we aimed to understand whether whole-brain network structure is related to this increased seizure likelihood. We used electroencephalography (EEG) to estimate brain networks from people with AD and healthy controls. We subsequently inserted these networks into a model brain and simulated disease progression by increasing the excitability of brain tissue. We found the simulated AD brains were more likely to develop seizures than the simulated control brains. No participants had seizures when we collected data, so our results suggest an increased probability of developing seizures at a future time for AD participants. Therefore functional brain network structure may play a role in increased seizure likelihood in AD. We also used the model to examine which brain regions were most important for generating seizures, and found that the seizure-generating regions corresponded to those typically affected in early AD. Our results also provide a potential explanation for why people with AD are more likely to have generalized seizures (i.e. seizures involving the whole brain, as opposed to 'focal' seizures which only involve certain areas) than the general population with epilepsy.

## Introduction

Alzheimer's disease (AD) is a neurological disorder characterised by pathological accumulation of amyloid-beta ($A\beta$) peptides and hyperphosphorylated tau protein in cortical tissue and neurodegeneration, resulting in progressive cognitive decline [1]. AD patients have a 6–10 fold increased risk of developing seizures compared to controls [2], with a prevalence of 10–22% [3] (although estimates have ranged from 1.5–64% [2, 3]). In rodent models, seizure phenotype has been related to hyperexcitable cortical tissue believed to be a consequence of AD pathology [4–9]. Understanding seizures in AD is crucial for developing novel treatments and a fuller understanding of both disorders, since the rate of occurrence of seizures are believed to be positively correlated with the rate of cognitive decline in AD [10–12].

A leading hypothesis for hyperexcitability in AD is that $A\beta$ deposition leads to neurodegeneration and abnormal hyperactivity including seizures, which in turn result in increased amyloid burden, leading to a self-amplifying neurodegenerative cascade [7]. In rodents, it has been observed that excessive neuronal activity can increase amyloid deposition [13, 14], while transgenic models of amyloidopathies often exhibit hyperexcitability [4–8] and synaptic degeneration [15, 16]. Computational modelling of this activity dependent degeneration has recreated alterations to electroencephalographic (EEG) recordings observed in humans with AD including slowing of oscillations and altered functional connectivity [17]. Similar effects were observed along with cortical hyperexcitability by targeting degeneration towards regions with high $A\beta$ burden in empirical PET recordings [18]. Tau pathology may also play a leading role in epileptogenesis in AD [3] in a similar cycle of deposition to the one described above, since evidence suggests that neuronal hyperactivity enhances propagation of tau [19] while excessive tau may increase local network excitability via stimulation of glutamate release [20, 21]. Tau levels may also mediate $A\beta$ toxicity and synaptic impairments [22, 23], suggesting that these mechanisms may be intertwined and that both amyloid and tau pathology may play a role in the increased prevalence of epilepsy in AD [3]. Additional key factors which may influence seizure likelihood in AD are vascular dysregulation, metabolic alterations and increased inflammation, resulting in neuronal activity dysregulation [24, 25].

While these hypotheses potentially explain increased excitability of local tissue in AD, evidence suggests the propensity of a brain to generate seizures is not only a result of local network excitability, but is also related to its large-scale functional network structure [26–31]. Alterations to large-scale functional network structure have widely been reported in AD based on studies from neuroimaging modalities including electroencephalography (EEG) [32, 33], magnetoencephalography [34–36], and functional MRI [37]. It is therefore possible that altered long-range functional connectivity in AD may contribute to increased susceptibility to seizures and, under the hypothesis of cyclical amplification of AD pathology and local excitability, facilitate the spread of pathological cortical hyperexcitability. Similarities have been observed between altered resting-state functional connectivity in humans with AD and epilepsy [3], consistent with this hypothesis. Furthermore, epilepsy patients with co-morbid AD have increased likelihood of generalized seizures than those without AD [3, 38], suggesting that large-scale connectivity is likely to play a role in seizure genesis in AD.

In this manuscript, we hypothesise that the large-scale functional networks of people with AD are more susceptible to seizures than those of cognitively healthy controls. To examine this hypothesis, we use the *brain network ictogenicity* (BNI) computational modelling framework [28, 39–41]. We assume that abnormal networks co-occur with increased cortical excitability for seizures to emerge in people with AD, and hence we analysed electrophysiological data in which functional network alterations have been observed in AD compared to controls [33] (none of whom experienced seizures). To understand the effect that these alterations might have on seizure generation, we used a mathematical model of seizure transitions in which cortical excitability was a free parameter [31, 40–43]. Our aim was to simulate an increase in cortical excitability in both healthy and AD brains and observe whether the concurrent abnormal network structure and increased excitability makes people with AD more likely to generate seizures *in silico* than controls. We also hypothesise that the regions primarily responsible for seizure generation in AD participants (as suggested by the computational model) correspond to those typically exhibiting high A$\beta$ burden [44, 45]. We test this hypothesis by calculating *node ictogenicity* (NI) [39–41], quantifies the degree to which a region governs susceptibility to seizures in the model.

## Materials and methods

### Ethics statement

All procedures were approved by the National Research Ethics Service Committee South West Bristol (Ref. 09/H0106/90). Participants provided written informed consent before participating and were free to withdraw at any time.

### Methodology

The methodology of the study is outlined in Fig 1. Source-space functional connectivity derived from the EEG was used to specify a network in a computational model of seizure transitions. To assess the susceptibility of the network to seizures, the excitability of cortical tissue was increased, and the fraction of time the simulated neural dynamics spent in the seizure state (called brain network ictogenicity, BNI) was calculated. The details of these calculations are described below.

### Data and functional networks

The current dataset has previously been analysed [33, 46], and pre-processing and functional network construction follow previously described methods [33]. Below, a brief overview of the

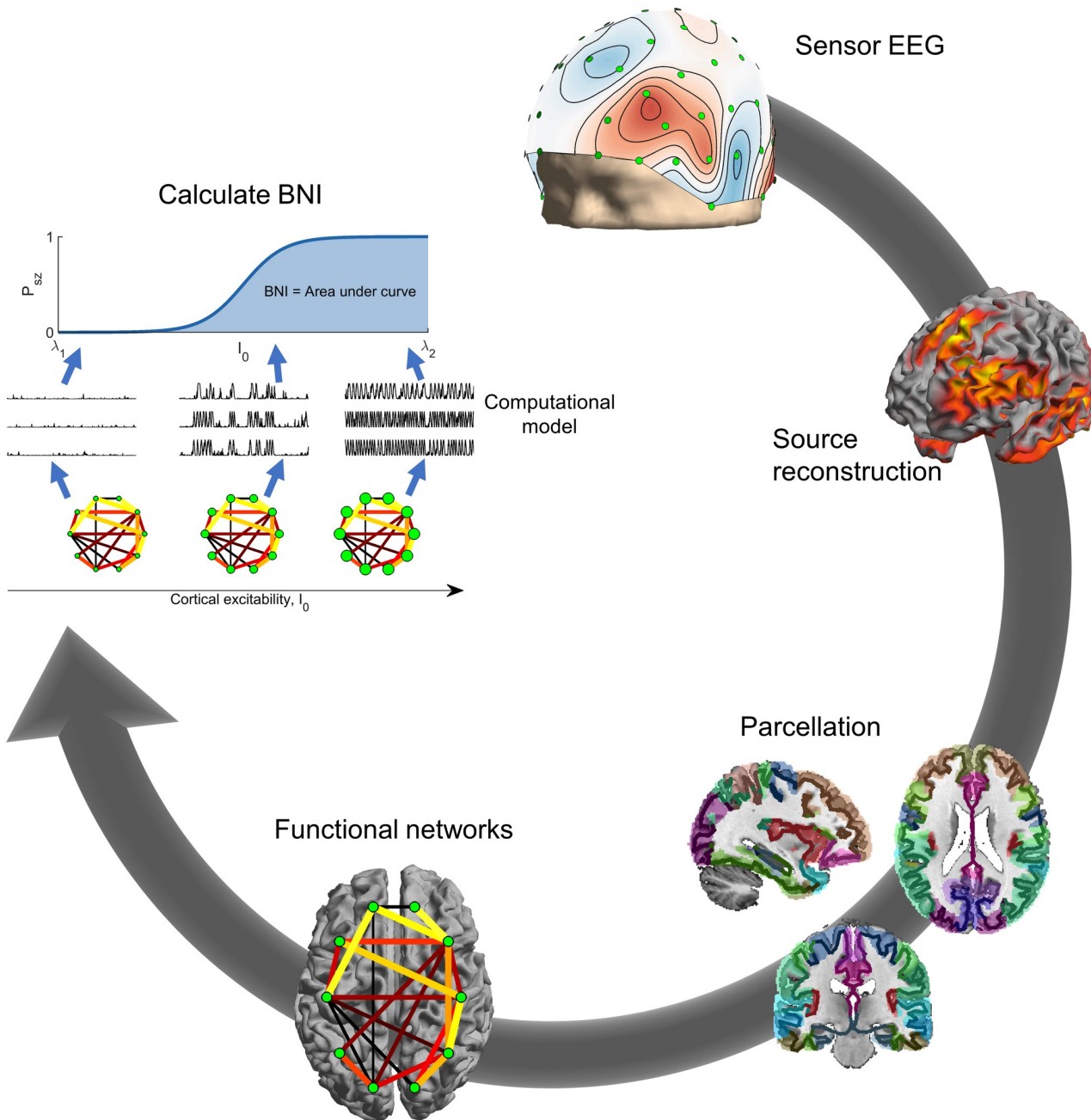

**Fig 1. Calculation of BNI.** Clockwise from top: Sensor EEG was source reconstructed using the eLORETA algorithm. The source solution was parcellated into 40 ROIs given by the Brainnetome atlas. Functional networks were calculated from the parcellated time courses of the 40 regions using theta-band phase locking factor. BNI was calculated by placing the network into a model of seizure transitions, and increasing the excitability of cortical tissue in the model. For each value of the excitability parameter $I_0$ (visualised by node size in the figure), the fraction of time spent in the seizure state by the simulated dynamics was calculated. BNI was the area under the curve of fraction of seizure time against $I_0$.

**Table 1. Participant demographics.** The columns showing age and mini-mental state examination (MMSE) scores show means and standard errors over participants.

| Cohort | Age (years) | MMSE | n | Male | Female |
|---|---|---|---|---|---|
| Controls | 76±7 | 29±1 | 26 | 14 | 12 |
| AD | 79±9 | 23±3 | 21 | 8 | 13 |

data and analysis pipeline are given. A very similar pipeline has been used to calculate functional networks for modelling BNI in source-space from scalp EEG in epilepsy patients [41], supporting the use of these methods for this study.

**Participants.** The cohort consisted of patients with a diagnosis of probable AD (n = 21, 13 female, 8 male) and age-matched cognitively healthy controls (n = 26, 12 female, 14 male). The AD group was recruited from memory clinics in the South West of England on a consecutive incident patient basis following clinical assessment. The diagnosis of probable AD was determined by clinical staff using the results of family interview, neuropsychological and daily living skills assessment according to DSM-IV [47] and NINCDS-ADRDA guidelines [48] together with neurological, neuroimaging, physical and biochemical examination. Age-matched controls were recruited from the memory clinics' volunteer panels; they had normal general health with no evidence of a dementing or other neuropsychological disorder, according to NINCDS-ADRDA guidelines [48]. All participants were free from medication known to affect cognition and had no history of transient ischemic attack, stroke, significant head injury, psychiatric disorder, or neurological disease with non-AD aetiology. All participants had no clinical history of seizures, but no extensive electrophysiological workup was performed to definitively rule out subclinical epileptiform activity [49, 50].

Participant demographics have previously been reported [33, 51, 52], and are given in Table 1. People with AD had significantly lower cognitive test scores than controls as assessed with the mini-mental state examination (MMSE), and there was no significant difference in age or sex between groups [52].

**EEG acquisition and pre-processing.** A single twenty second, eyes-open resting-state epoch of 64-channel EEG sampled at 1 kHz was analysed per participant. Visual and cardiac artifacts were manually rejected using independent component analysis, and data was bandpass filtered at 1–200 Hz, demeaned, detrended, and re-referenced to average using the Fieldtrip toolbox [53].

**Source reconstruction.** The Fieldtrip toolbox was used for source reconstruction. For all participants, we used a template forward model implemented in Fieldtrip. The source-model was the canonical cortical surface implemented in Fieldtrip consisting of 5124 dipoles distributed along the cortical sheet. Dipoles were oriented normal to the surface [54, 55]. The volume conduction model was Fieldtrip's template 3 layer boundary element method model [56]. Template head models have been demonstrated to perform well compared to individual models derived from MRI [57].

Source reconstruction used exact low resolution electromagnetic tomography (eLORETA) [58, 59], which is a linear, regularized, weighted minimum norm estimate with zero localization-error. eLORETA is suited to the study of whole-brain phase synchronization [60, 61], analysis of resting-state data [46, 62], and source-spaced modelling of BNI from scalp EEG [41].

The 5124 dipole source-space was parcellated into 40 regions of interest (ROIs) based on the Brainnetome atlas [63] by assigning each ROI the time course corresponding to the first principal component of dipoles within that ROI [33]. The time course of the first principal component of all voxels in the ROI is a single time series whose value at each time point is

minimally different to the activity of all voxels, i.e. it accounts for a maximal spatial variance. For the list of ROIs, see S1 Table.

**Functional networks.** Computation of functional networks used in this study followed previously described methods [33]. Time courses of ROIs were bandpass filtered into the theta-band (4–8 Hz), and the Hilbert transform used to estimate instantaneous phases. Functional networks were constructed by calculating the phase locking value (PLV) [64] between the filtered time courses of pairs of ROIs. Potentially spurious edges were rejected based on a null distribution of PLV values constructed from 99 pairs of iterative amplitude adjusted Fourier transform surrogates [65]. PLV values that did not exceed 95% significance vs the surrogates were rejected. Furthermore, to reduce the likelihood of spurious connections due to source leakage, PLV values with zero phase-lag were rejected. Zero-phase lag here corresponds to a mean phase difference smaller than the phase resolution at 4 Hz, given the sampling rate, i.e. if the mean phase difference was less than $(2\pi \times 4)/f_{sample} = 0.008\pi$ radians, the edge was set to zero. The Dijkstra algorithm was used to compute the length of the shortest path between all pairs of nodes, and edges with an indirect shortest path (i.e. the shortest path is not the single edge between the pair of nodes) were also rejected [27]. Surrogate-corrected PLV derived from resting-state EEG have been shown to be useful for BNI modelling in both sensor- [28, 42] and source-space [41] in patients with epilepsy.

## Computational model of seizure transitions

**Computational model.** To test the hypothesis that altered network structure and increased cortical excitability makes people with AD more prone to develop seizures than healthy controls, we used a phenomenological model of seizure transitions in which we could control cortical excitability, namely the theta-model [31, 40–42, 66], a phase oscillator model where stable phases represent resting brain activity and rotating phases represent seizure activity (see Fig 1 in [40]). Each ROI is described by a phase oscillator $\theta_i$ whose activity is given by

$$\dot{\theta}_i = (1 - \cos\theta_i) + (1 + \cos\theta_i)I_i(t). \tag{1}$$

$I_i(t)$ is an input current received by ROI $i$ at time $t$,

$$I_i(t) = I_0 + \sigma\xi^{(i)}(t) + \frac{K}{N}\sum_{i \neq j} a_{ji}[1 - \cos(\theta_j - \theta^{(s)})], \tag{2}$$

which comprises the excitability $I_0$ of the ROI, noisy inputs $\xi^{(i)}(t)$ from remote brain regions, and the interaction of ROI $j$ connected to $i$ as defined by the adjacency matrix $A = (a_{ji})$ (i.e. the PLV values of the functional network). $K$ is a global scaling constant weighting network interactions relative to cortical excitability and noise. $N$ is the number of ROIs. $\theta^{(s)}$ is a stable phase to which the oscillators converge in the absence of noise and interaction (see e.g. [40] for more details). A phase oscillator is at rest if $I_i(t) < 0$ or rotating if $I_i(t) > 0$. The transition at $I_i(t) = 0$ corresponds to a saddle-node on invariant circle (SNIC) bifurcation (see Fig 1 in [66]).

For simplicity, we assumed that all ROIs had the same cortical excitability $I_0$ and consequently the same $\theta^{(s)}$. The noise $\xi^{(i)}(t)$ was modelled as Gaussian noise with zero mean and unit standard deviation, with noise magnitude $\sigma = 6$ as in previous studies [31, 40–42]. Simulations used a stochastic Euler method with time step $\delta t = 10^{-2}$ (arbitrary units) and total integration time $T = 4 \times 10^6$. All parameters used for simulations and their descriptions are given in Table 2.

**Brain network ictogenicity (BNI).** We are interested in the effect of increasing $I_0$ on the propensity of a network to generate seizures. To quantify this seizure susceptibility, we used the concept of brain network ictogenicity (BNI) [28, 31, 39, 40]. First, we defined the average

**Table 2. Parameters, their meanings, and their standard values used in the simulations.** All parameters have arbitrary units.

| Parameter | Meaning | Value |
|---|---|---|
| $I_0$ | Excitability of ROIs | Range [-1.7,0.5] |
| $K$ | Global coupling strength | 10 |
| $N$ | Number of ROIs | 40 |
| $A$ | Connectivity matrix between regions | PLV from data |
| $\theta^{(s)}$ | Stable steady state in absence of noise or connections | $\cos^{-1}\left(\frac{1+I_0}{1-I_0}\right)$ |
| $\sigma$ | Standard deviation of noise | 6 |
| $T$ | Total number of simulation steps | $4 \times 10^6$ |
| $\delta t$ | Time step for simulation | $10^{-2}$ |

proportion of time spent in seizures, $P_{sz}$, for a given $I_0$ as

$$P_{sz}(I_0) = \frac{1}{N}\sum_{i=1}^{N}\frac{t_{sz}^{(i)}(I_0)}{T},$$

(3)

where $t_{sz}^{(i)}(I_0)$ is the time that ROI $i$ spends in the rotating phase (i.e. in the seizure state) during a simulation time $T$ (we used $T = 4 \times 10^6$ (arbitrary units) as in previous studies [41, 42]). $P_{sz}(I_0)$ is in the range zero (no seizures) to one (always in the seizure state). We computed the BNI as

$$\text{BNI} = \int_{\lambda_1}^{\lambda_2} P_{sz}(\lambda)\ d\lambda,$$

(4)

where the range $[\lambda_1, \lambda_2]$ was chosen so that all brain networks assessed had $P_{sz}$ varying from zero to one. Increasing $I_0$ results in increasing the input currents of all the oscillators in the network, which in turn makes them more likely to rotate. Our hypothesis is that networks from people with AD may require a lower $I_0$ for their $P_{sz}$ to be higher than 0 than healthy people. Consequently, we expect the BNI from people with AD to be higher than the BNI from healthy people, since the inflection point of the BNI curve would occur for smaller values of $I_0$. For the comparison between the two groups to be meaningful, the BNI was computed using the same parameters $K$, $\lambda_1$ and $\lambda_2$ for all participants.

**Node ictogenicity (NI).**   Each ROI has its own unique set of connections to other ROIs implying that each ROI may have a different contribution for the network's ability to generate seizures. To measure the contribution of each ROI to BNI, we computed the node ictogenicity (NI) [39, 40]. The calculation of NI consists of measuring the BNI upon the removal of a ROI from the network to infer the ROI's importance for the generation of seizures. The NI of ROI $i$ is given by

$$\text{NI}^{(i)} = \frac{\text{BNI}_{\text{pre}} - \text{BNI}_{\text{post}}^{(i)}}{\text{BNI}_{\text{pre}}},$$

(5)

where $\text{BNI}_{\text{pre}}$ is the BNI before removing ROI $i$, whereas $\text{BNI}_{\text{post}}$ is the BNI after removing ROI $i$ (and all its connections) from the network.

NI can be interpreted as follows. If node $i$ has no influence on seizure generation, then there will be no change in BNI following removal of the node and hence $\text{BNI}_{\text{post}}^{(i)} = \text{BNI}_{\text{pre}}$ and $\text{NI}^{(i)} = 0$. Conversely if node $i$ is entirely responsible for seizure generation in the network,

then seizures are suppressed completely following the removal of the node, and hence $\mathrm{BNI}_{\mathrm{post}}^{(i)} = 0$ and $\mathrm{NI}^{(i)} = 1$. In most real cases, removal of an ROI will reduce seizure propensity but not completely suppress seizures, and hence $0 < \mathrm{BNI}_{\mathrm{post}}^{(i)} < \mathrm{BNI}_{\mathrm{pre}}$ and $0 < \mathrm{NI}^{(i)} < 1$, where larger values indicate seizures are more suppressed following removal of the node. A negative value of $\mathrm{NI}^{(i)}$ suggests that this node suppresses seizures, and hence removal of the node increases seizure propensity (i.e. $\mathrm{BNI}_{\mathrm{post}}^{(i)} > \mathrm{BNI}_{\mathrm{pre}}$).

To assess group averages it is convenient to further define a normalised NI (nNI),

$$\mathrm{nNI}^{(i)} = \frac{\mathrm{NI}^{(i)}}{\sum_{j=1}^{N} \mathrm{NI}^{(j)}}, \tag{6}$$

which preserves the relative importance of each ROI for seizure generation, while removing potential differences in absolute NI values between different networks.

## Statistical analysis

All statistical analysis used non-parametric measures, which do not rely on assumptions about the distribution of the data. All pairwise comparisons used the Mann-Whitney U test, for which we report the $U$-statistic and its $z$-score [67] as a measure of effect size of the changes. For paired group-level statistics we use Friedman tests and report $\chi^2$ as a measure of effect size. For testing which ROIs contribute most significantly towards the generation of seizures, we use a null hypothesis that all nodes contribute equally, and use a non-parametric bootstrap to calculate a null distribution under this null hypothesis. Specifically, if the empirical data is represented as $N_{participant} \times N_{ROIs}$ nNI values, we generate 10,000 surrogate nNI data sets with the same dimensions but with entries bootstrap-sampled from the original data. This destroys any effect of ROI on the nNI distribution. We then compare the median (over participants) nNI value for each ROI against the same statistic from the surrogates to obtain a $p$-value, which is then controlled for multiple comparisons using the Benjamini-Hochberg false discovery rate procedure.

For comparisons of NI/nNI distributions between groups, multi-variate pattern analysis (MVPA) was performed with the MVPA-Light toolbox [68], using the spatial distributions of NI/nNI as features. Classification used logistic regression, with the 5-fold cross-validated area under the ROC curve (AUC) as the performance metric. 20 repetitions of this procedure were performed and the average AUC was used in subsequent analysis (i.e. statistical testing via permutation analysis, described below). The AUC metric is reported as mean ± standard deviation across folds and repetitions. Permutation testing was used to assess significance of differences between groups, following the same methodology as the original data (e.g. the same cross-validation folds and number of repetitions). No regularization or feature selection was used to reduce overfitting, and hence MVPA classification rates may not be robust or generalizable to new populations of data. However, each permutation used the same cross-validation folds and hence the degree of overfitting is equal in the original and permuted data sets. Therefore any difference in AUC between permuted and empirical data suggests that there is an association between NI/nNI distribution and disease group in the empirical data which is not present after permutation.

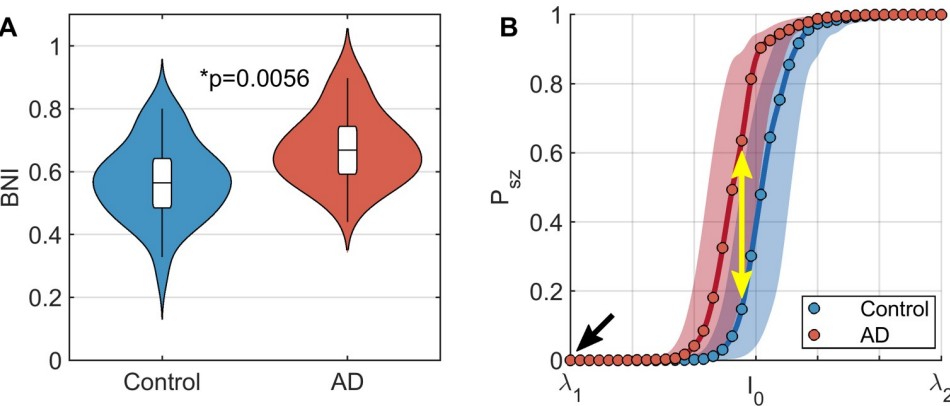

**Fig 2. Functional brain networks in AD are more susceptible to seizure generation in response to increase cortical excitability than controls.** (A) Violin plots of BNI in people with AD and controls. BNI is significantly higher in AD. (B) Plots of seizure likelihood, $P_{sz}$, against excitability, $I_0$. Lines show median values over all participants within a group, while shaded regions show interquartile ranges. Circles show the grid of values on which $I_0$ was simulated. The values of BNI shown in subplot A are the area under these curves for each participant. The black arrow shows a hypothetical 'current' baseline, in which no participants have seizures. The yellow arrow shows that if cortical excitability increases, AD participants are more likely to experience seizures than controls. Parameters: $K = 10$, $\lambda_1 = -1.7$, and $\lambda_2 = -0.5$.

## Results

### Elevated brain network ictogenicity in AD

We first tested whether brain networks in people with AD had a higher propensity to generate seizures than controls by quantifying BNI. BNI was calculated as the area under the curve of percentage of time spent in seizure as cortical excitability ($I_0$) is increased. The lower bound for $I_0$, which we term $\lambda_1$, was chosen to be a value at which no participant exhibits seizures in the simulation. This reflects the fact that at the time of EEG acquisition no participants exhibited seizures. Therefore baseline excitability in the model, i.e. $I_0 = \lambda_1$ (black arrow in Fig 2B), represents a non-seizure state for all participants.

We found that BNI was significantly larger for AD than controls ($U = 403$, $z = 2.7710$, $p = 0.0056$; Fig 2A, with median BNI curves shown in Fig 2B). Hence, as we increase cortical excitability in the model, AD patients develop seizures for smaller values of $I_0$ than controls. Since the only individual differences in the model are the functional brain networks, this suggests AD brain networks are more susceptible to seizure generation. A consequence of this result is that for a given level of cortical excitability (e.g. the yellow arrow in Fig 2B), an AD simulation is statistically more likely to have seizures than controls.

The global coupling constant $K$ is a free parameter of the model. For the results shown in Fig 2, we used $K = 10$. S1 Fig shows that the results are consistent for other values of $K$. For the remainder of the analysis, we therefore focus on $K = 10$.

### Spatial distribution of regions responsible for seizures in AD simulations

Having identified that brain networks from AD participants have a greater propensity to generate seizures than controls, we next studied which ROIs are responsible for emerging seizures in the simulations for these patients. To do this, we calculated the NI of each ROI in the network, which quantifies the importance of ROIs for simulated seizure generation by quantifying the reduction in seizures after removing the ROI from the network. To avoid weighting results

more strongly towards participants with higher total NI, we normalised NI distributions to unit sum for each participant, i.e. we used nNI values (Eq 6).

We first tested whether there were 'seizure driving' ROIs in the AD participants, i.e. whether certain ROIs had consistently higher nNI across AD participants than others, and therefore the distribution of nNI was not homogeneous over the cortex. A Friedman test found this to be the case, since nNI score significantly depended on ROI ($\chi^2 = 75.87$, $p = 3.69 \times 10^{-4}$). We therefore subsequently examined which ROIs contributed most to seizure generation in the AD simulations. Fig 3A shows the distribution of nNI values over the cortex. To test the degree to which different regions deviate from the null hypothesis of homogeneously distributed nNI, we used a non-parametric bootstrap test (see Materials and methods). The null median nNI scores were normally distributed (S2 Fig), so for visualization of deviation from the null distribution we show $z$-scores for each ROI against the surrogate distribution in Fig 3B. The bilateral cingulate, orbital, and fusiform cortices had the largest deviations, with bilateral cingulate, right fusiform, and left orbital exceeding the 5% significance level against the null distribution (Benjamini-Hochberg corrected non-parametric bootstrap). Interestingly, as a group-level observation we note that these regions seem to be consistent with those with the largest and earliest staged A$\beta$ burdens in AD [44, 45], but this was not

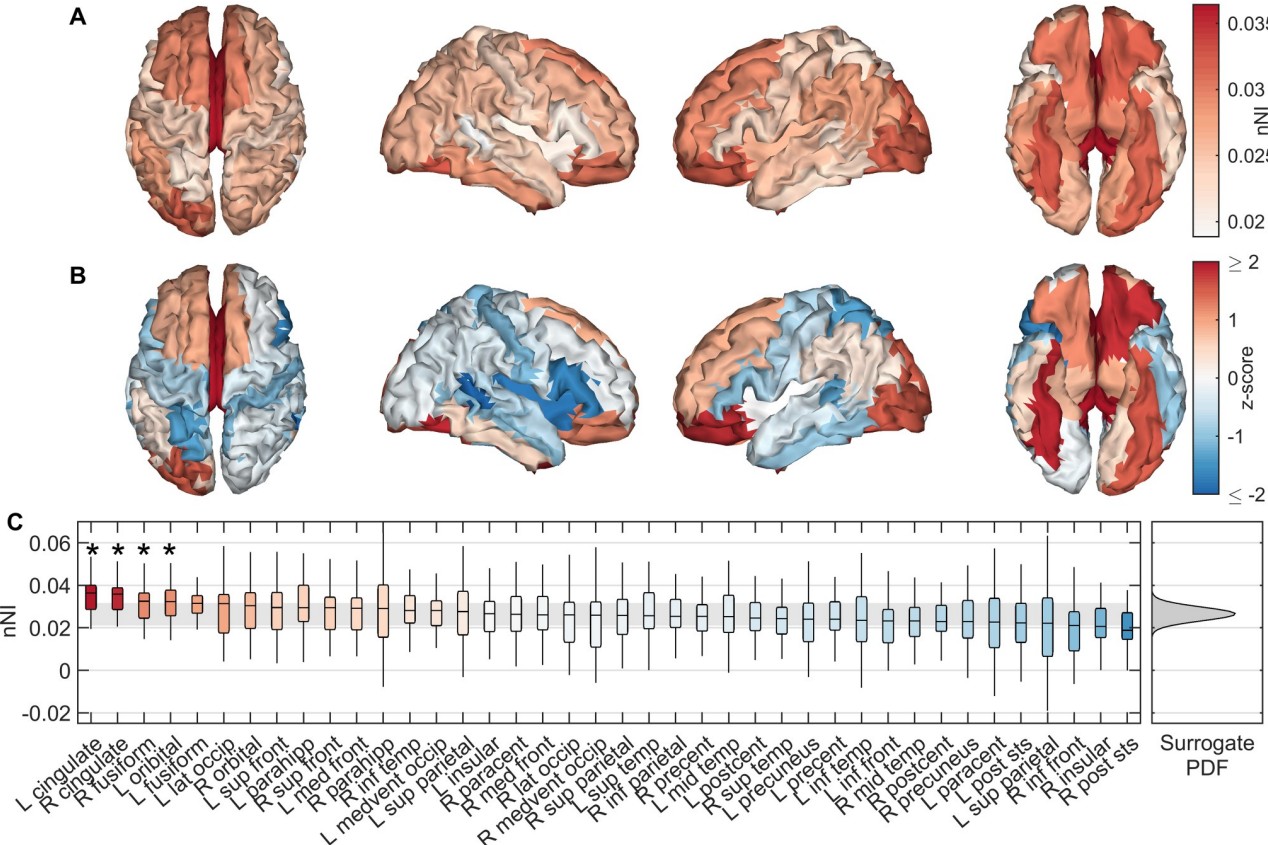

**Fig 3. Spatial distributions of seizure generating regions in AD simulations.** (A) Median nNI over participants for each ROI. (B) $z$-scores of median nNI against the surrogate distribution. Red scores suggest the nNI score was larger than expected from a homogeneous distribution, suggesting these regions are most strongly responsible for generating seizures. (C) nNI values for regions sorted by median over participants by descending nNI. Background colour shows $z$-score. The shaded grey region shows empirical 95% confidence intervals on the surrogate data, while the full (Gaussian) probability density function of the surrogate data is shown on the right. ROIs marked by an asterisk were significant to (FDR corrected) $p < 0.05$. Full names of ROIs along with the abbreviations given here are given in S1 Table. Parameters are those given in Fig 2.

tested statistically on the individual-level as no amyloid PET scans were available for our participants.

## Comparison of node ictogenicity with controls

Cognitively healthy participants may also develop epilepsy, so it is of interest to examine whether the most likely ROIs to be responsible for seizure generation in our model are consistent between the control and patient groups. Here we compare the spatial distribution of NI values in people with AD against controls.

Fig 4 shows differences in NI distributions between groups. Both mean ($U = 141$, $z_U = -2.81$, $p = 0.0049$, Mann-Whitney U test; Fig 4A) and standard deviation ($U = 149$, $z_U = -2.64$, $p = 0.0082$, Mann-Whitney U test; Fig 4B) of NI were significantly lower in the AD participants. We next examined whether the spatial patterns (as opposed to global statistics such as mean and standard deviation) differed between groups (Fig 4C). To do so, we used multivariate pattern analysis (MVPA), treating NI scores at each ROI as features. MVPA demonstrated a significant difference in the spatial distributions of NI (AUC = $0.7231 \pm 0.165$, $p = 0.0060$). However, since MVPA identified no significant differences in nNI distributions (which controls for mean NI) between AD and controls (AUC = $0.5220 \pm 0.165$, $p = 0.4080$), differences between groups in spatial patterns were primarily due to the decrease in mean NI in AD rather than different spatial topographies of NI values.

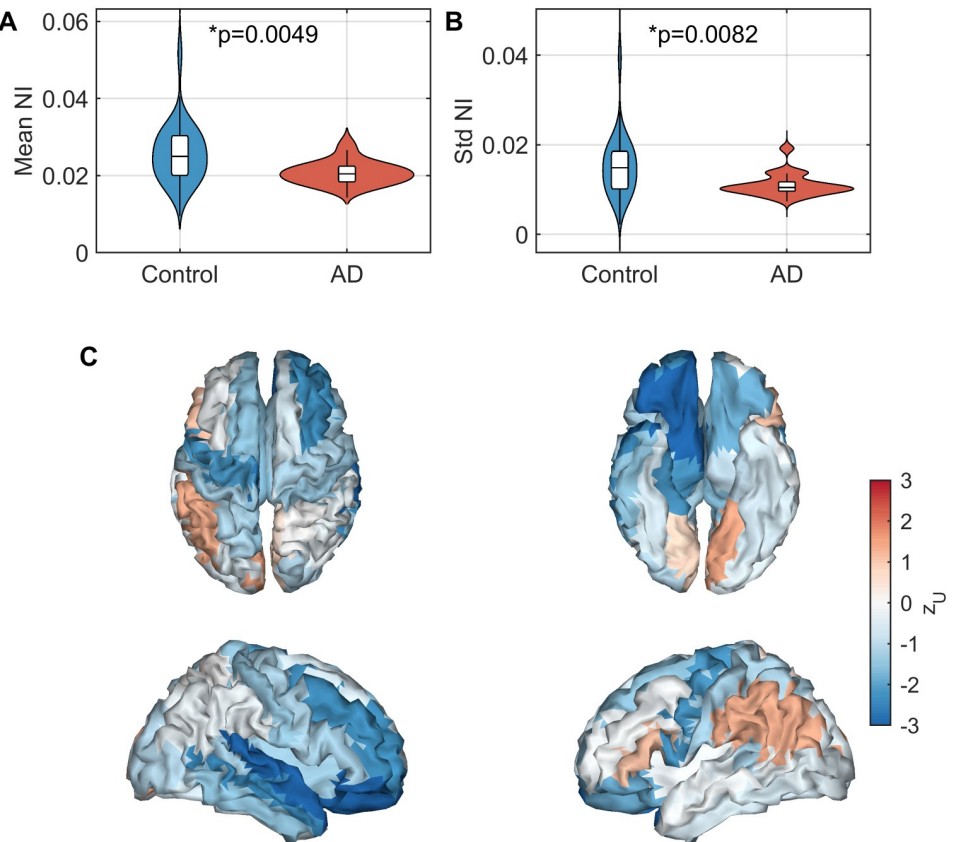

**Fig 4. Analysis of NI scores in AD relative to controls.** (A) Mean NI across nodes. (B) Standard deviation of NI across nodes. (C) Effect sizes of differences in NI between AD and controls for each ROI, quantified by the *z*-score of the *U*-statistic. Parameters are those given in Fig 2.

## Discussion

In this manuscript, we used a computational model of seizure transitions in brain networks [40] to examine the potential relationships between alterations to large-scale functional network structure [33] and increased prevalence of seizures in AD.

At present, most conceptual models for development of seizures in people living with AD focus on the mechanisms of increased local excitability of cortical tissue [3, 7]. However, large-scale functional network structure also likely plays a crucial role in determining the propensity of a brain to generate both focal and generalized seizures [26–31, 69]. A key result of our study is that previously reported alterations to functional connectivity in AD [33] result in brain networks which more readily generate seizures in response to increased cortical excitability than cognitively healthy controls. This was quantified using the *brain network ictogenicity* (BNI) framework [28, 39–41].

This result fits closely with the self-amplifying cascade hypothesis for epileptogenisis in AD, which suggests that AD pathologies induce hyperexcitability [4–9], while the resulting hyperactivity drives an increase in the pathologies [13, 14]. For a review of this cascade hypothesis, see the Introduction, [7], [3], and references therein. While at present this is an untested hypothetical model, under this hypothesis we can interpret increased BNI in AD as follows. At baseline, all participants have a level of cortical excitability such that no seizures are observed, here modelled by setting $I_0$ equal to $\lambda_1$ for all participants (black arrow in Fig 2). As initial amyloid/tau deposits form, cortical excitability is increased. The AD participants are more likely to develop seizures than controls as a result of this increased excitability (yellow arrow in Fig 2). The increased seizure propensity is a direct consequence of the altered functional network structure, as this is the only difference between AD and control simulations in our study. This hyperactivity mediates an increase in A$\beta$ and tau burden [13, 14], which in turn may amplify excitability [4–9]. Future work should involve testing this hypothetical self-amplifying cascade within our model framework. It would be particularly interesting to longitudinally track the evolution of BNI throughout cognitive decline, as it is currently unclear whether seizure likelihood evolves with disease progression.

To quantify the importance of an ROI for ictogenesis in our model, we removed the ROI and recalculated BNI. The resulting change in BNI is termed the node ictogenicity (NI) for that ROI [39–41]. This was repeated for all ROIs, to calculate a distribution of local ictogenicity. Interestingly, the cingulate, fusiform, and orbital cortices had greatest NI (Fig 3). [44] developed a neuropathological staging of AD related changes in the brain based on postmortem analysis, finding that orbital and medial temporal (including fusiform) regions were the earliest affected by A$\beta$ pathology, while deposits in the cingulate regions appeared in the 2nd stage. [45] recently performed amyloid-PET scans to develop an *in vivo* staging of A$\beta$, placing cingulate, inferior temporal, and fusiform cortices in the earliest stage, while orbital cortex was one of the earliest affected in stage 2. In our model, therefore, the ROIs which have the potential to most strongly drive seizures are those stereotypically found to have the earliest and strongest A$\beta$ burden in AD. These same regions are also those affected by tau burden at middling stages (Braak stages III-IV) of the disease [44]. Furthermore, [70] developed an *in vivo* staging of cortical atrophy in AD, and in the earliest stages reported 20–30% cortical grey matter loss in the medial temporal, posterior cingulate, and orbitofrontal cortices, which similarly align to the regions with largest NI in our model. However, a 15–20% loss of grey matter in the temporoparietal region was additionally reported which does not correspond to our results. These associations support a group-level relationship between seizure propensity and regions stereotypically affected by AD at the earliest stages, which should be tested on an individual-level using multimodal imaging in future work. When distributions of NI were controlled for

individual mean effects, there was no difference in the NI distribution between AD and controls. A potential interpretation of this result in our model is that the primary difference between AD and control participants is that the cascade of excitability vs pathology happens more quickly in AD due to large-scale network structure, and not that certain regions are more strongly targeted than others e.g. as a result of pathology. Future work should examine this hypothesis.

Another interesting finding from the local analysis was lower mean NI and spatial standard deviation of NI distributions in AD participants than controls (Fig 4). Lower mean NI in AD suggests that, on average, removing a single node from the brain network will be less likely to suppress seizures in our simulations than in controls, and hence more distributed groups of nodes are likely to be responsible for driving seizures. Lower standard deviation of NI distributions in AD suggests that there is more spatial homogeneity in the importance of nodes to drive seizures than in controls (heterogeneity suggests some nodes play a key role in driving seizures while other nodes have very minor role). Combined, these results are suggestive of a generalized (as opposed to focal) mechanism for seizures in AD. For focal seizures, one might expect the seizure onset zones to have higher NI than other ROIs and removal of these foci to drastically reduce BNI, while other ROIs may be less influential, resulting in a high spatial variance in NI scores [31, 69]. In contrast, decreased variability in the importance of nodes for generating seizures combined with an overall decreased mean NI suggests that ictogenicity is more homogeneously distributed across the cortex in people with AD than controls, which in turn may imply that people with AD are more susceptible to generalized seizures than controls. In the general population of people living with epilepsy, generalized onset tonic-clonic seizures are the main seizure type in approximately 10% of cases [71], while for an elderly population with transient amnestic epilepsy (i.e. epilepsy with interictal transient amnestic dementia-like symptoms without an AD aetiology) this prevalence is as low as 4% [3, 72]. Conversely, in people with AD and other dementias with comorbid epilepsy, the prevalence of generalized onset tonic-clonic seizures is 15–40% [3, 73–76]. These reports are therefore in line with our results, and therefore large-scale brain network structure is likely an important factor in determining seizure phenotype in AD.

## Methodology

There are several methodological considerations to this study, as functional network structure is likely to be dependent on analysis pipeline and influences the BNI/NI results [40]. Here, we chose *a priori* a single pipeline that was appropriate to for the scientific question at hand. The methods used for construction of functional connectivity were derived from a previous study [33] which showed differences in functional connectivity between controls and AD using graph theoretical metrics. A similar pipeline was additionally used in a previous source-space study for BNI analyses demonstrating usefulness for the model-based ictogenicity analysis [41].

We analysed a single 20 second epoch of resting-state EEG per participant. Studies of phase locking have demonstrated reliable estimates can be made with as little as 12 second segments of data [77], while the PLV has shown high test-retest reliability between recording sessions [78]. However, it is also known that over periods of several hours or days there are fluctuations in functional connectivity statistics [79–81], so future studies could examine whether BNI measures differ within subjects from different recording segments/sessions.

Other methodological choices may influence the resulting functional network. These included the choice of frequency band [82], source reconstruction algorithm [61], brain atlas [83] including number of nodes [84], and functional connectivity metric [85] used to construct

the functional network. The methodology used to construct the networks used in this study was discussed in detail in previously work [33], but will be touched upon briefly here. The alpha frequency band has been chosen for several studies using computational models to assess seizure likelihood from functional connectivity [27, 42, 69, 86]. These studies were all performed with eyes-closed data, where the alpha band dominates the EEG. However, this study used eyes-open data, in which the alpha network is less powerful and differs in functional network structure [87–89]. Therefore here the theta-band was used, motivated by past studies which have demonstrated theta-band alterations to EEG functional connectivity in AD [33] and epilepsy [27, 90–92], and a relationship between theta-band dynamics and cognitive impairment in dementia [17, 33, 93]. eLORETA was used for source reconstruction. eLORETA has been demonstrated to outperform other source reconstruction algorithms for resting-state data [46, 61, 62] and is suitable for phase synchronization [60, 61], particularly in studies with a similar number of electrodes (60–71) to the one presented here [60–62]. eLORETA has also been shown to be useful for computational modelling of BNI in source space [41].

We used phase locking value (PLV) to calculate functional connectivity. A key limitation of PLV is that it may be influenced by leakage in the source reconstruction solution. To minimize the effects of leakage, we used a low resolution atlas consisting of only 40 ROIs [33] and set to zero any PLV values that had on average zero phase-lag [27, 33]. This methodology is likely to be less conservative than the use of leakage-correction schemes such as orthogonalization of source time series [94, 95] or metrics such as the phase lag index [96] or the imaginary part of coherence [97]. In spite of potential influence of leakage, PLV has been demonstrated to be a powerful tool for source functional connectivity analysis. Simulation studies have demonstrated that PLV (in the absence of leakage correction) can accurately capture functional connectivity in the source space solution [61, 98, 99], and has high within-subject consistency between recording sessions [78]. Furthermore, PLV is a useful measure of large-scale connectivity for simulating seizure dynamics [28, 41, 42, 69, 86]. These results justify our use of PLV in this study.

## Limitations and future work

All participants in this study had no history of seizures at the time of data acquisition, and while AD patients are 6–10 times more likely to develop seizures than controls [2], there is no guarantee that our AD participants will develop seizures while the controls will not. Future work should additionally introduce two cohorts of people with seizures—those comorbid with AD and those without AD. While the work presented here potentially gives insights into the network-level mechanisms of increased seizure prevalence in AD, the comparison between people with AD who develop seizures and those who don't would help further elucidate the specific network mechanisms which contribute to seizure propensity in AD. Another key factor which should be examined in future work is APOE genotype, a known risk factor for both Alzheimer's disease and epilepsy [100, 101]. However, it is likely that later stage AD participants or people with epilepsy will be treated by potentially EEG-altering pharmacological interventions; one key advantage to this study of early stage AD participants is that all participants were free from medication at the time of data acquisition.

In this work, we identified spatial distributions of regions with high seizure propensity in AD, and observed a resemblance to stereotypical patterns of A$\beta$ pathology in AD patients. A limitation to further exploring this observation and quantifying statistical effects is the absence of amyloid PET scans in our cohort. Future work should involve integration of multi-modal neuroimaging data, including functional data (EEG/MEG/fMRI), structural MRI, and PET, to

quantify the relationship between spatial patterns of NI and Alzheimer's pathologies, cortical atrophy, vascular dysregulation, metabolic alterations, inflammation, etc. on an individual level.

A limitation of the modelling methodology is the use of a static functional network which is independent of cortical excitability. This separation of local and network mechanisms has been shown to be informative in previous applications to epilepsy [27, 39, 40, 43, 86]. It lies between a standard functional connectivity analysis on the one hand, and the full inversion of a biophysical model on the other. The latter would simultaneously estimate intrinsic excitability of nodes and connectivity between them, thereby capturing the role of local dynamics in shaping large-scale functional connectivity [102]. However, reliable estimation of the large number of parameters in such a model is challenging.

In addition, future work involving more biophysically realistic modelling could incorporate activity dependent degeneration [17], in which both the local dynamics and large-scale connectivity between populations are altered along the simulated disease progression. Our modelling also assumes homogeneity of local dynamics across regions of the brain, which is also likely to be of limited biophysical realism. Integration with other imaging modalities such as PET/MRI would allow for spatial heterogeneity in local excitability related to statistics such as amyloid/tau burden, vascular/metabolic dysfunction, inflammation, or neurodegeneration.

## Conclusions

In this study we have demonstrated potential large-scale brain network mechanisms for increased seizure propensity in people living with Alzheimer's disease. In a computational model in which functional connectivity was the only subject-specific parameter, AD participants were more likely to develop seizures than healthy controls in response to an increase in excitability of cortical tissue. No patients in this study had seizures at the time of EEG acquisition, so results should be interpreted as a group-level probability of developing seizures at a future time. Examination of ROIs necessary for seizure generation in the model uncovered two main findings. Firstly, the most important ROIs for seizure generation were those typically burdened by A$\beta$ at the early stages of AD. Secondly, alterations to the large scale network structure in AD potentially play a role in determining seizure phenotype, namely an increased likelihood of generalized seizures in AD. Future work should involve contrasting seizure-free AD participants with co-morbid AD/epilepsy participants, as well as integration of multimodal neuroimaging data and biophysically realistic modelling to gain further insight into the mechanistic relationships between regional seizure propensity and AD pathology.

## Supporting information

**S1 Fig. Results are consistent across values of K.** The plots on the left and centre recreate Fig 2 for a range of values of global coupling constant *K*. The correlation matrix on the right shows Spearman's correlation of BNI scores across participants as different values of K are used. All correlations were $\geq$ 0.9665.
(TIF)

**S2 Fig. Median nNI scores from bootstrapped samples of AD nNI scores are not significantly different from a normal distribution.** Probability density function (pdf; left) and cumulative distribution functions (cdf; right) for the empirical data and best fit normal distribution. A Kolmogrov-Smirnov test showed no significant differences between the empirical and normal distributions ($p$ = 0 using Matlab's kstest function).
(TIF)

**S1 Table. A list of ROIs for parcellation of source data, based on the coarse grained Brainnetome atlas [63] used in Tait et al. (2019) [33].** For each ROI, we give a full name, and the abbreviation used in Fig 3.
(PDF)

## Author Contributions

**Conceptualization:** Luke Tait, Marinho A. Lopes.

**Data curation:** George Stothart, Nina Kazanina.

**Formal analysis:** Luke Tait, Marinho A. Lopes.

**Investigation:** Luke Tait, Marinho A. Lopes.

**Methodology:** Luke Tait, Marinho A. Lopes.

**Resources:** George Stothart.

**Software:** Luke Tait, Marinho A. Lopes.

**Supervision:** Jiaxiang Zhang, Marc Goodfellow.

**Visualization:** Luke Tait.

**Writing – original draft:** Luke Tait.

**Writing – review & editing:** Marinho A. Lopes, George Stothart, John Baker, Nina Kazanina, Jiaxiang Zhang, Marc Goodfellow.

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
