## [Decision Letter · Decision Letter 0]

1 Apr 2021

Dear Dr Tait,

Thank you very much for submitting your manuscript "A Large-Scale Brain Network Mechanism for Increased Seizure Propensity in Alzheimer's Disease" for consideration at PLOS Computational Biology.

Let me apologize for the delay in this first decision, I hope you can appreciate that these are challenging times for many, and committing and sticking to a time schedule can be more problematic.

As with all papers reviewed by the journal, your manuscript was reviewed by members of the editorial board and by several independent reviewers. In light of the reviews (below this email), we would like to invite the resubmission of a significantly-revised version that takes into account the reviewers' comments.

The issues are of a twofold nature: some details about the method and the results need to be specified, and the method itself needs to be better inscribed in the state of the art, in particular regarding the actual insight on the mechanism. If the technical issues are addressed, but doubts still remain on this latter aspect, we might offer publication in PLOS One.

We cannot make any decision about publication until we have seen the revised manuscript and your response to the reviewers' comments. Your revised manuscript is also likely to be sent to reviewers for further evaluation.

Sincerely,

Daniele Marinazzo

Deputy Editor

PLOS Computational Biology

Daniele Marinazzo

Deputy Editor

PLOS Computational Biology

Reviewer's Responses to Questions

**Comments to the Authors:**

Reviewer #1: The authors perform a relatively simple and straightforward analysis using (1) functional connectivity from resting state EEG data from AD and controls previously published and (2) an existing model for seizure prediction and ictogeneity previously used in epilepsy studies. The manuscript is well written and the work is transparently presented. While there are clear limitations to this study (for ex seizures or epileptiform activity or clinical follow-up not being assessed in these subjects) I believe this paper does add something new to the existing literature of AD. Although there are various papers evaluating hyperexcitability, seizures and epileptiform activity, here the authors bring a different perspective by evaluating the impact of long range connections to seizures.

The value of this paper is introducing a mechanism from the epilepsy field to the AD field, involving functional connectivity and its organization. This concept becomes clearer in the results and discussion, but it not very clear in the abstract. At least I felt that the abstract states that the authors are proposing a mechanism but what is this mechanism is unclear. Maybe this could be stated more clearly in the abstract, dedicating 1-2 sentences since is this where the meat of the message is.

This work is a preliminary test of a hypothesis/mechanisms, and the authors highlight in the discussion and limitations that their dataset is limited so that future studies with a more suited dataset should assess reproducibility and clinical relevance. If that is the goal, it would be important that this work produces a package (meaning methodological description or code) that can be more readily picked up by a future researcher: so that a future researcher can directly apply the same methods than the authors in this manuscript and test their hypothesis. For example the following would be useful:

- Sharing more details so that the analyses can be reproduced by other researchers (particularly on how to compute the NI and BNI values for each network). The main formulas of the theta model and BNI are presented but some details seem to be missing. How was the simulation performed, what values were used for all the parameters of the model

- How was the analysis implemented? Custom scripts, toolbox, reference scripts from other researchers, etc?

Line 214: the text states: “At baseline excitability (black arrow in Fig 2B), no participant exhibits seizures. This is in line with the observation that no patients studied here exhibit seizures”.(and similar sentence in line 286) Why is that? Does baseline excitability mean Io=lambda1? How was lambda1 selected? Why would lambda1 represent the normal stage of AD subjects and why would the fact that the patients do not have seizures be related to lambda1? How would this look like for an epilepsy patient?

Interpretation of NI and nNI values: What range of values can NI have and what do they mean?

Within AD, Increased nNI in regions typically associated to AD pathology, interpreted as these being the regions that drive seizures

However there are decreased NI values AD vs control. Does this mean that AD nodes are less prone to drive seizures than controls? How does this fit together with the previous point

Why is the variability deviation of NI higher in controls than in AD? Given that the AD subgroup should be a more heterogeneous group, in particular when it comes to excitabilty and seizures, shouldn’t the variability in AD be higher? In the discussion the authors indicate that this could be driven by less spatial variability in AD than in controls, and this would seem a valid explanation when inspecting variability or STD across brain regions. However, figure4 A shows the average across all brain regions, so in principle region-to-region variability should not have an influence on the values. What could be the explanation?

Paragraph starting 281. Authors are describing a hypothesis of the pathophysiology of AD, excitability and epileptogenesis. It is good to share such a hypothetical model in such a transparent way for the reader, but it should perhaps be clear at the start that this paragraph is stating an unproven hypothetical model. Additionally, the authors seem to be describing a model where excitability and seizure probability increase progressively across disease progression (moderate dementia subjects would have more probability of seizures than MCI subjects), but, is that really the case?

The authors seem to be implying that their AD subjects have no seizures and may develop seizures in the future, but they are “normal” at the time of EEG acquisition. However, various studies indicate that AD studies without known seizures may have substantial epileptiform activity, especially during night, but that one would not notice unless one does long full night EEG recordings or even better implanted electrodes in the medial temporal lobe to have enough sensitivity to inspect local medial temporal lobe activity. If the AD patients included in this study have no clinical history of seizures but have not undergone an in depth evaluation, it does not seem unlikely that some could have seizures / epileptiform activity that have not been detected.

References:

https://onlinelibrary.wiley.com/doi/abs/10.1002/ana.24794

https://n.neurology.org/content/95/16/e2259.abstract

Methods for PLV calculation. Line 129 states “ PLV values with zero phase-lag were rejected. Edges with a stronger indirect path were also rejected”. The phase lag will rarely be exactly zero so I imagine there are some thresholds to assess this. Could you share more details on this? Also on the direct and indirect paths are computed and compared.

Reviewer #2: In this paper Tait et al explore reasons for increased occurrence of epileptic seizures in people with Alzheimer’s disease. To this end they recorded 20s multichannel EEG of patients with Alzheimer’s disease (n=21) to age matched controls (n=26). Using source localisation the recorded signals were mapped to 40 cortical ROIs of the Brainnetome atlas. All signals attributed to the same ROI were combined into an ‘eigen’-signal based on the first PCA component. Connectivity between pairs ROIs was then based on the theta phase-locking value (PLV). Leading to connectivity graphs at rest for AD patients and controls. The resulting adjacency matrices are used in a computational model of seizure transitions. The model allowed to quantify the time spent in seizures based on the network and the excitability of a region of interest (I0). A range of I0 values (from \\lambda_1 to \\lambda_2) was used, and for each value the fraction of time spent in seizures was computed. From this, the brain network ictogenicity (BNI) is defined as the area under that curve. The BNI was compared between AD subjects and controls and the authors found that BNI was larger for AD than for controls. Further, by omitting a node from the network and comparing the full network BNI to the BNI w/o the node they quantified the influence for each node (i.e., region of interest): the node ictogenicity (NI) or normalized NI (nNI). An MVPA comparison of regional features between cases and controls showed that the differences between groups are mainly due to global reductions in NI compared to regional specific reductions.

Overall the paper is clearly written and easy to follow; limitations have been discussed in sufficient detail. The authors rely on their tried and tested techniques of BNI and NI to study this cohort of AD subjects and controls. And while the modeling and statistics part of the paper is very clear, there are some aspects that require further clarification (see below).

Major:

1. The authors use age and sex matched controls. The sex-matching, however, is not perfect (14 males in controls vs 8 males among case). Statistically, the sex difference was not significant. However, sex differences in connecivity have been reported across different imaging modalities - sometimes being introduced as artifacts from brain morphology. Thus, the sex-imbalance might influence the results here as well. (e.g., bias brain morphology on source localization or genuine sex differences in connectivity).

I suggest to conduct a supplementary analysis with a subsable with perfect sex matching to assess whether results remain significant (or if failing that, at least show the same dirction/pattern).

2. In the MVPA approach, Logistic Regression is used. Was there any regularization used? There were 40 regions of interest (ROIs) and 49 samples, not making for an overly robust analysis, especially given that 5-fold CV was carried out, thus providing ~40 samples per training fold. Moreover, could the authors clarify the phrase “20 repetitions of this procedure were performed and the best performing repetition was used in subsequent analysis” . it is not clear what the subsequent analysis is (next part is the results section). Also selecting the best run out of 20 repetitions appears to be some sort of overfitting. In addition, reporting AUC as mean and SE across CV folds is fine, but also across repetitions is leading to artificially small confidence intervals. Since SE depends on the sample size - and one can arguably run just more repetitions to get the SE as small as desired.

3. The authors state themselves that one “limitation of the modelling methodology is the use of a static functional network which is independent of cortical excitability.” In fact, this reviewer has the feeling that the analysis became detached from the original question about cortical excitability and seizures. This is overall a very, very evolved analysis and the only important parameter that differs between cases and controls is the adjacency matrix. This leads to the question of whether the results should be interpreted in terms of excitability and seizure potential.

Overall, many studies have demonstrated different functional connectivity profiles between AD and controls - predominantly around the default mode network - across a wide array of network derived properties. How can the authors ensure that the observed change of BNI is not simply related to one of the classic network properties?

To be more convincing in this regard, the work would need some verifiable predictions (e.g., “people with highest BNI did actually evolve seizures”), or at least some form of specificity analysis e.g., compare the findings here to another analysis between controls and patients with another brain disorder (with aftropy) but that does not exhibit seizures. Is the BNI here as well different between cases and controls? Other options could be, e.g., among controls to compare men and women or test the association between BNI and age - in these analysis one would not - per se - expect differences in BNI.

Minor:

1. In formula (2) the index appears to be incorrect: \\theta_i should be \\theta_j .

Reviewer #3: This a very interesting study, focused on modeling/understanding the increased tendency of the AD brain to develop seizures. The authors used functional brain networks estimated from EEG data and generative computational modeling to simulate brain activity, where a given parameter allows to explore how the healthy control and AD brains will behave in terms of seizure generation. The study, and its findings, are of interest for the research community. I have, however, a few major comments that should be clarified before considering the manuscript for publication:

1) The procedure to construct individual functional brain networks from EEG data involve multiple empirical steps. For example, after source reconstruction, (i) filtering the signal to the theta band, (ii) removing connections not reaching the 95% significance, (iii) removing connections with strong indirect paths, etc. Each of these steps, can imply or cause obtaining different networks for NC and AD subjects (differences in EEG-frequency have been reported, strength of connections have also been documented, indirect effects may appear more often in one population than the other, etc). If the networks used as input on the phenomenological model are very different, it would not be surprising at all the obtained differences in seizure tendencies for NC and AD groups, which would appear just as a direct result of network reconstruction procedures (i.e. may result from artificial effects). The authors need to explore/clarify the impact of their network-reconstruction steps on the global and regional ictogenicity analyses. In addition, if the NC and AD networks would be matched in terms of number of connections (and their location), would the observed results still valid?

2) Since the Introduction, the authors made an emphasis on the link between amyloid and tau deposition with the increased seizure tendency in AD. First, the are many other well-documented biological alterations that can be causally related to such effects. For example, vascular dysregulation, metabolic alterations and increased inflammation, resulting in neuronal activity dysregulation. The authors should also comment on those potential effects, as a way to make their study less biased towards a unique pair of biological factors (mostly towards amyloid). Avoiding an amyloid-centered and -limited view will significantly increase the value of the study.

3) In relation with the previous comment, the link between increased regional ictogenicity in AD and amyloid deposition seems forced. The authors are finding a few areas that, yes, are among those usually identified with high amyloid burden. But, are the whole brain regional patterns similar? Can this be statistically tested? Amyloid deposition in the brain is very unspecific and widespread, almost all brain regions get amyloid at some point. This is different for tau, structural atrophy, and, as said, many other biological factors involved in AD pathomechanisms. In addition to comparing with whole brain regional amyloid patterns, could the authors compare with at least two other factors altered in AD?

**Have the authors made all data and (if applicable) computational code underlying the findings in their manuscript fully available?**

Reviewer #3: **No: **It is not explicitly clear if the data and code are available.

PLOS authors have the option to publish the peer review history of their article (what does this mean?). If published, this will include your full peer review and any attached files.

Reviewer #1: No

Reviewer #2: No

Reviewer #3: **Yes: **Yasser Iturria Medina

**Have all data underlying the figures and results presented in the manuscript been provided?**

Reviewer #1: None

Reviewer #2: Yes
---

## [Decision Letter · Decision Letter 1]

28 Jun 2021

Dear Dr Tait,

Thank you very much for submitting your manuscript "A Large-Scale Brain Network Mechanism for Increased Seizure Propensity in Alzheimer's Disease" for consideration at PLOS Computational Biology. As with all papers reviewed by the journal, your manuscript was reviewed by members of the editorial board and by several independent reviewers. The reviewers appreciated the attention to an important topic. Based on the reviews, we are likely to accept this manuscript for publication, providing that you modify the manuscript according to the remaining review recommendation by reviewer 1.

Sincerely,

Daniele Marinazzo

Deputy Editor

PLOS Computational Biology

Daniele Marinazzo

Deputy Editor

PLOS Computational Biology

[LINK]

Reviewer's Responses to Questions

**Comments to the Authors: **

Reviewer #1: Review included as pdf attachment

Reviewer #2: Thank you for this thorough revision. I only have one minor comment. Please used 'sex' instead of 'gender' when refering to demographics (line 99) since this is in reference to the biology.

[see: https://www.nature.com/articles/s12276-019-0341-0]

Reviewer #3: The authors have clearly replied my comments, with valid points/results. I don't have further comments and recommend the manuscript for publication.

**Have the authors made all data and (if applicable) computational code underlying the findings in their manuscript fully available?**

Reviewer #1: Yes

Reviewer #2: Yes

Reviewer #3: None

PLOS authors have the option to publish the peer review history of their article (what does this mean?). If published, this will include your full peer review and any attached files.

Reviewer #1: No

Reviewer #2: No

Reviewer #3: No

Figure Files:

Data Requirements:

Reproducibility:

References:

---

## [Editor Report · Decision Letter 2]

6 Jul 2021

Dear Dr Tait,

We are pleased to inform you that your manuscript 'A Large-Scale Brain Network Mechanism for Increased Seizure Propensity in Alzheimer's Disease' has been provisionally accepted for publication in PLOS Computational Biology.

Before your manuscript can be formally accepted you will need to complete some formatting changes, which you will receive in a follow up email. A member of our team will be in touch with a set of requests. Please include the link to the code repository both in the paper and in the data availability statement.

Best regards,

Daniele Marinazzo

Deputy Editor

PLOS Computational Biology

Daniele Marinazzo

Deputy Editor

PLOS Computational Biology

---

## [Editor Report · Acceptance letter]

30 Jul 2021

PCOMPBIOL-D-21-00297R2 

A Large-Scale Brain Network Mechanism for Increased Seizure Propensity in Alzheimer's Disease

Dear Dr Tait,

I am pleased to inform you that your manuscript has been formally accepted for publication in PLOS Computational Biology. Your manuscript is now with our production department and you will be notified of the publication date in due course.

With kind regards,

Andrea Szabo
